# Delay Discounting as a Potential Therapeutic Target for Weight Loss in Breast Cancer Survivors

**DOI:** 10.3390/cancers14051134

**Published:** 2022-02-23

**Authors:** Jasmine S. Sukumar, Jennifer E. Vaughn, Allison Tegge, Sagar Sardesai, Maryam Lustberg, Jeffrey Stein

**Affiliations:** 1Division of Medical Oncology, The Ohio State University Comprehensive Cancer Center, Columbus, OH 43221, USA; jasmine.sukumar@osumc.edu (J.S.S.); sagar.sardesai@osumc.edu (S.S.); 2Stefanie Spielman Comprehensive Breast Center, The Ohio State University, Columbus, OH 43212, USA; 3Division of Hematology, Department of Internal Medicine, The Ohio State University Comprehensive Cancer Center, Columbus, OH 43210, USA; 4Center for Health Behaviors Research, Fralin Biomedical Research Institute at VTC, Virginia Tech, Roanoke, VA 24016, USA; ategge@vt.edu (A.T.); jstein1@vtc.vt.edu (J.S.); 5Department of Statistics, Virginia Tech, Roanoke, VA 24060, USA; 6Division of Medical Oncology, Yale Comprehensive Cancer Center, New Haven, CT 06510, USA; maryam.lustberg@yale.edu; 7Department of Human Nutrition, Foods, and Exercise, Virginia Tech, Roanoke, VA 24061, USA

**Keywords:** delay discounting, breast cancer, obesity, survivorship, behavioral health

## Abstract

**Simple Summary:**

Obesity is a rising health epidemic in breast cancer survivors and associated with multiple negative health sequalae and increased mortality. Delay Discounting (DD) is a behavioral economic measure of an individual’s valuation of future outcomes. While higher DD correlates with obesity in the general adult population, valuation of the future may impact cancer survivors differently due to their unique experiences. We assessed cross-sectional associations between DD, BMI, and healthy lifestyle behaviors in an exploratory analysis of 89 women with hormone receptor positive non-metastatic breast cancer. We found higher DD to be associated with obesity and decreased frequency of vegetable consumption. Future studies should investigate DD as a therapeutic target for novel behavioral interventions in breast cancer survivors affected by obesity. This may improve valuation of the future, increase healthy lifestyle behaviors, and facilitate weight loss to promote overall health and longevity in this population.

**Abstract:**

Obesity in breast cancer (BC) survivors is associated with increased mortality. Delay discounting (DD) is a behavioral economic measure of how individuals value future outcomes. Higher DD correlates with obesity in the general population. Valuation of the future may be associated with obesity differently in cancer survivors. This study evaluated the relationship between DD and obesity in BC survivors. We report an exploratory analysis assessing cross-sectional associations between DD, BMI, and lifestyle behaviors (vegetable and fruit consumption, exercise) related to obesity in 89 women with hormone receptor positive non-metastatic BC. Multivariate linear regression analysis examined demographic and lifestyle behavior variables associated with both BMI and DD. Greater willingness to wait for larger, delayed rewards (lower DD) was significantly associated with lower BMI (standardized beta = −0.32; *p* < 0.01), independent of age, race, income, time since diagnosis, and menopausal status. There was no significant association between DD and fruit consumption or exercise frequency. Vegetable consumption was significantly associated with lower DD (standardized beta = 0.24; *p* < 0.05). Higher DD is associated with obesity and decreased frequency of vegetable consumption in BC survivors. Future studies should investigate DD as a therapeutic target for behavioral interventions to facilitate weight loss and promote longevity in this population.

## 1. Introduction

World-wide, breast cancer (BC) is the most commonly diagnosed cancer in women, with an estimated 2.3 million new cases in 2020 [1]. Outcomes are significantly improving in early stage BC, with an approximate five-year survival rate of 90% [2], leading to a growing population of greater than 3.8 million women in the United States who are BC survivors [3]. In addition to the risk of cancer recurrence and mortality, BC survivors also face other unique challenges associated with treatment such as obesity, depression, and a rising incidence of cardiovascular disease which can negatively impact long-term health [4]. Optimization of modifiable cancer risk factors through encouragement of healthy lifestyle behaviors is at the cornerstone of consummate cancer survivorship care.

Obesity is a rising health epidemic in the United States [5] and excess adiposity is a known risk factor for BC development. Additionally, there is an increased risk of BC recurrence and mortality associated with excess body weight in BC survivors [6,7,8,9]. For example, in a meta-analysis of 82 studies including 213,075 breast survivors, a 41% increase in mortality was demonstrated in women who were obese at the time of cancer diagnoses [8]. Obesity in BC survivors is also correlated with other adverse health sequelae, including an increased risk of second primary malignancies [10], and negative impact on multiple quality of life factors, such as chronic fatigue, sexual dysfunction, body image, lymphedema, and neuropathy [11]. Additionally, there is an increased risk of chronic medical conditions known to be associated with increased adiposity such as cardiovascular disease and insulin resistance with resulting diabetes [11,12]; the presence of these comorbidities is a substantial cause of shortened survival in this population. Weight loss is, therefore, a highly impactful therapeutic target in BC survivorship. However, weight loss interventions in BC survivors have been associated with mixed outcomes in success and suboptimal adherence [13,14,15]. Novel approaches to improve long-term patient engagement and adherence of healthy lifestyle choices through identification of basic decision-making processes and intervention with behavioral modification may promote durable weight loss and its resulting long-term health benefits.

Delay discounting (DD) is a behavioral economic measure that describes the degree to which individuals value future outcomes [16]. When given a choice between a larger reward in the future or a smaller more immediate reward in the present, many individuals will prefer the latter, particularly as the delay to the more beneficial outcome increases [17]. Individuals with higher, or more rapid, DD rates therefore have an increased bias for immediate gratification and a greater devaluation of future outcomes [16,18]. For example, a person with a high rate of DD may be more likely to choose to eat a delicious, but unhealthy meal now, rather than consider an option which could positively impact future fitness and health. High DD rates are implicated across many maladaptive health behaviors [19,20,21,22] in the behavioral science literature, including those related to obesity, such as sedentary behavior [23], poor glycemic control [24,25], and poor dietary choices (consumption of obesogenic foods and increased caloric intake) [26,27]. Predictably, higher DD rates are correlated with obesity [27,28]. Thus, consistent with the experimental medicine approach to behavior change research [29], DD may serve as a therapeutic target in obesity; that is, interventions that reduce DD may, in turn, facilitate weight loss. Such methods to engage DD as a target for weight loss are being implemented as an emerging treatment approach [30,31,32,33], but there is a paucity of data regarding the association between DD, obesity, and behavioral choices among those who have experienced a cancer diagnosis.

Valuation of the future in cancer survivors may be altered by a number of unique factors, such as an adjusted mortality perception, financial toxicity from cancer diagnoses and treatment, and cancer-related psychosocial consequences. For instance, cancer survivors may experience psychologic effects including posttraumatic stress and a fear of cancer recurrence [34,35]. The latter may be prevalent in nearly all cancer survivors and can persist despite completion of therapy and considerable time since diagnoses [35,36]. Moreover, a traumatic experience and the resulting negative impact on overall well-being may produce a sense of foreshortened future and perception of early mortality [37]. An individual’s discount rate may also be impacted by distinct cancer-specific factors, such as diagnosis, stage, treatment history, and time since diagnosis. For example, in a study of cancer survivors of multiple cancer types, time since cancer diagnoses was negatively associated with DD rate (*p* = 0.01), suggesting that DD is elevated soon after diagnosis and decreases with time [38]. The multitude of these complex factors may all impact valuation of the future and potentially decrease an individual’s commitment to long-term goals and rewards related to health.

Focused investigation of DD in cancer survivors is critical to better elucidate its role as a potential therapeutic target for weight loss in this population. In a study of 1001 survivors across several types of cancers, including breast, lung, sarcoma, genitourinary, and others, participants were recruited to assess associations between DD and multiple lifestyle behaviors [38]. Lower DD rates were associated with several healthy lifestyle behaviors (lower alcohol consumption, no cigarette or other tobacco use, no tanning bed use, attending annual primary care visits), highlighting the applicability of this behavioral principle to cancer survivorship. Notably, no significant interaction was found between DD rate and Body Mass Index (BMI), or between DD rate and physical activity or vegetable consumption. An important limitation of this study in specific regard to the association between DD and BC was the heterogeneous cohort of survivors, as different cancers are known to be associated with different prognoses and unique survivorship goals. A minority of patients were BC survivors (n = 175, 17.5%) and approximately half of the sample was of male gender. Notably, there was no investigation of whether cancer type could be a predictor of the interaction between DD and health behaviors. Thus, focused assessment of DD and lifestyle behaviors in BC survivors is warranted to better ascertain the relationship to adiposity, nutrition, physical activity, and specific cancer characteristics (e.g., stage, time since diagnoses).

A pilot study from our investigational team was the first to study DD specifically in a BC survivor cohort and demonstrated DD as a predictor of adjuvant endocrine therapy non-adherence in 89 participants. This study found that treatment adherence was higher in participants showing greater willingness to wait for larger, delayed rewards (i.e., lower DD) (standardized beta = 0.328, *p* = 0.005) [39]. These findings suggest that DD does impact lifestyle behaviors in BC survivors similar to other populations and strategies to reduce DD may be efficacious to improve long-term survival. Herein, we describe additional exploratory analyses in this cohort to investigate the cross-sectional association between DD, BMI, and healthy lifestyle behaviors associated with energy balance. We hypothesized that increased BMI and associated negative lifestyle behaviors in BC survivors are associated with higher DD, similar to that observed in individuals without cancer [23,26,27,28]. Such findings could prompt further, longitudinal studies of this association, and ultimately, the development of novel behavioral strategies for weight loss in this population to address an unmet clinical need in BC survivorship.

## 2. Materials and Methods

### 2.1. Study Design and Eligibility

Patients were recruited from a large, community-based healthcare system in the Roanoke, VA area. Key eligibility criteria included: (1) women with stage 1–3 non-recurrent hormone receptor positive invasive BC treated with curative intent in the last five years, (2) currently prescribed or recommended adjuvant oral endocrine therapy (tamoxifen, anastrozole, letrozole, or exemestane) with or without ovarian suppression, and (3) 18–80 years old. Patients who were prescribed endocrine therapy for metastatic disease and those suffering from physical (e.g., non-ambulatory) or cognitive (e.g., dementia) impairments that may interfere with medication self-administration were excluded. All participants completed informed consent prior to study enrollment. The Carilion Clinic Institutional Review Board approved all study procedures.

The primary aim of this study was to evaluate cross-sectional associations between DD and endocrine therapy non-adherence. We previously reported these findings which demonstrate that greater rates of DD were significantly associated with poorer adherence to endocrine therapy [39]. To assess the relationship between DD and obesity, we undertook several exploratory analyses, which we describe herein: to evaluate the cross-sectional correlation between (i) DD and BMI, (ii) BMI and positive lifestyle behaviors, and (iii) DD and positive lifestyle behaviors.

Weight and height were collected by self-report. BMI was calculated based on the formula: BMI = weight (kg)/height (m)^2^. During a single 90–120 min session, participants completed a questionnaire to collect information encompassing demographics, cancer diagnoses and treatment, and healthy lifestyle behaviors. The latter included information on consumption of fruit and vegetables (number of servings per day; six categories: ranging from less than one serving a day up to five or more servings a day) as well as exercise frequency (moderate intensity physical activity; six categories: never, less often than once a year, a few times a year, a few times a month, a few times a week, daily or almost daily). Next, participants completed the five-trial, adjusting-delay task [39] to assess DD of both $100 and $1000 (order randomized). This method has been validated and is accepted in the behavioral science literature as a reliable means by which to measure DD [27,28,33]. Two amounts were examined because DD varies with the amount of reward [18]. Thus, analyses of multiple amounts provide more generalized estimates of choice. In this task, participants made repeated, hypothetical choices between a larger amount ($100 or $1000, depending on task iteration) available after a delay and half of this amount ($50 or $500) available immediately. Across trials, the delay to the larger amount is titrated based on previous choices until reaching an indifference delay (possible range: 1 h–25 years, in approximately logarithmic intervals), at which point the subjective value of both options is approximately equal. The indifference delay in this task serves as a measure of Effective Delay 50 (ED50), or the delay required for the larger reward to lose 50% of its value [39,40]. The dependent measure of DD was ED50 (in days). Longer ED50 values reflect less discounting (i.e., greater willingness to wait). ED50 values were non-normally distributed (positive skew) and were natural log transformed prior to analysis.

### 2.2. Statistical Analyses

Demographic variables are summarized using means (standard deviations), medians (interquartile ranges), and frequencies (percentages), where appropriate. Univariate linear regression analysis was performed to identify explanatory variables, including demographics and healthy lifestyle behaviors (exercise frequency, fruit and vegetable consumption), associated with BMI. To identify an optimal subset of factors associated with BMI, we used multivariate regression and performed an exhaustive model selection search and identified the final model as that with the lowest Bayesian Information Criterion (BIC). Similarly, multivariate regression with model selection were performed to predict discounting (ED50 $100 and ED50 $1000). Results are reported using standardized betas and *p*-values. A *p*-value < 0.05 was considered significant in this study.

## 3. Results

A total of 89 participants completed this study, of which details have been previously published [39]. Table 1 describes the baseline demographic, lifestyle, and clinical characteristics of the cohort. The mean age at diagnosis was 58.7 years (range: 33–77) and the mean time since diagnoses was 2.5 years. The majority (n = 58, 65%) of participants were post-menopausal and of white race (n = 85, 96%). Most individuals were with stage 1 (n = 39, 44%) or stage 2 (n = 25, 28%) BC. The mean BMI was 29.8 kg/m^2^ (SD 6.6). A total of 29 (33%) patients in this cohort were overweight; 38 (43%) patients were obese. Adjuvant aromatase inhibitor (anastrozole, letrozole, or exemestane) was the most common endocrine therapy prescribed (n = 53, 60%), with the remainder receiving adjuvant tamoxifen (n = 36, 40%).

For DD, the mean natural log ED50 was 5.38 (1.89) and 5.69 (2.14) for ED50 $100 and $1000, respectively. For context, these values correspond to raw values of 217.02 and 295.89 days, respectively, indicating that the $100 and $1000 rewards lost half of their subjective value in approximately 7–10 months. High ED50 values (i.e., lower DD) were associated with lower BMI (ED50 $100, standardized beta = −0.32, *p* = 0.002; ED50 $1000, standardized beta = −0.28, *p* = 0.008) and this effect was independent of age, race, income, time since diagnosis, and menopausal status (Figure 1). The median frequency of healthy lifestyle behaviors included 3 (IQR 2–4) vegetable servings/day, 3 (IQR 2–3) fruit servings/day, and exercise a few times a week (IQR a few times a month to daily or almost daily). Exercise frequency (standardized beta = −0.36, *p* < 0.001), fruit consumption (standardized beta = −0.32, *p* = 0.002), and vegetable consumption (standardized beta = −0.24, *p* = 0.024) were significantly associated with lower BMI. Model selection identified the most probable model to include ED50 $100 (standardized beta = −0.25, *p* = 0.009), exercise frequency (standardized beta = −0.28, *p* = 0.006), and fruit consumption (standardized beta = −0.22, *p* = 0.029).

Linear regression models were used to estimate ED50 from health behaviors. Exercise frequency (ED50 $100, standardized beta = 0.13, *p* = 0.242) and fruit consumption (ED50 $100, standardized beta = 0.15, *p* = 0.164) were not associated with DD rate; however, vegetable consumption was significantly associated with lower DD rate (ED50 $100, standardized beta = 0.24, *p* = 0.026). Upon model selection, only age (ED50 $100, standardized beta = 0.28, *p* = 0.004) and household income (ED50 $100, standardized beta = 0.38, *p* < 0.001) were associated with DD rate.

## 4. Discussion

In this study of women with non-metastatic hormone receptor positive BC, we found a significant cross-sectional association between elevated DD and increased BMI in an exploratory analysis. This relationship is consistent with previously reported findings in the general adult population affected by increased BMI [27,28]. Thus, the association between DD and obesity persists despite the unique challenges faced by individuals with a cancer diagnosis. However, it contrasts the findings of Sheffer and colleagues, who did not observe a significant association between these variables in a mixed cohort of cancer survivors [38]. This suggests that the effect of DD may be different amongst cancer survivor populations. Particularly, diagnoses of distinct cancer types may impact future thinking on choice differently. This may be related to considerable variations in prognoses, symptom burden, and quality of life based on specific cancer diagnoses. For example, BC patients have a higher cure rate and live longer compared with individuals with many other solid tumor malignancies of a similar stage [2]. Thus, research on behavioral mechanisms underlying decision making in cancer patients may benefit from studies that either focus on individual cancer diagnoses or examine a potential moderating role of diagnosis type.

If the association between DD and obesity in BC survivors is confirmed in longitudinal studies, DD may be engaged as a therapeutic target, and both established and investigational behavioral therapeutic interventions have demonstrated success to decrease DD rate [42,43]. For example, one promising behavioral intervention is episodic future thinking (EFT), derived from the emerging science of prospection. In EFT, individuals generate and simulate personally meaningful and detailed future events to increase valuation of the future. The goal is to shift temporal orientation to the present [44]. EFT has successfully reduced DD rate in obese individuals [31,33,45,46,47], leading to effects of negative energy balance, including decreased caloric intake, diminished consumption of energy-dense foods, improved diet quality, and weight loss [30,31,32,33,45,48,49,50]. For example, Sze et al. observed that in an EFT study in obese participants, delivered by smartphone, there was a favorable decrease in energy intake and reduction in weight in adults following the four-week intervention compared with control [30]. Additionally, a brief EFT intervention implemented in participants of the present study demonstrated that DD is also amendable to reduction in BC survivors as compared to a control condition [39]. Based on these findings and the preliminary data presented in this report, implementation of behavioral interventions such as EFT to target DD for weight loss and promotion of positive lifestyle behaviors may also hold promise as a clinical tool in cancer survivors. These interventions may be adapted in future studies, with the goal to improve impactful clinical endpoints related to obesity in cancer survivors.

This study is not without limitations. The results of this cross-sectional analysis provide insight into potential mechanisms underlying increased weight in BC survivors, but cannot determine causality. Furthermore, the objectives studied were exploratory and additional, prospective research specifically powered to assess for these effects in a cohort of BC survivors with obesity is warranted. Longitudinal assessment of adiposity collected serially would also provide a more comprehensive understanding of obesity and DD following a cancer diagnosis. Namely, changes in weight and body composition following a cancer diagnosis is not uncommon and can have important clinical implications. These additional data could inform the optimal design of behavioral interventions to target DD for weight loss. Our investigative team is currently evaluating the prospective effects of DD and related targeted interventions on weight loss after a breast cancer diagnosis (NCT05012176). In addition, we did not find a consistent, statistically significant difference between DD rate and healthy lifestyle behaviors assessed in this study. While increased vegetable consumption was associated with lower discounting, there was no significant relationship between discounting and the other lifestyle behaviors. This may be related to the small sample size. Additionally, the potential to introduce recall bias exists with the use of self-report questionnaires. Future studies should incorporate validated, higher-resolution measures to provide a more comprehensive understanding of the relationship between discounting, physical activity, and energy consumption. For example, portion size, calorie consumption, diet quality, and exercise intensity/time are measures which can provide greater precision to estimate net energy balance and may clarify the link between discounting and lifestyle behaviors. Therefore, further investigation to assess the role of physical activity and dietary factors as a potential mediator of the association between DD and obesity should be pursued using robust measurements (e.g., fitness tracking, food frequency questionnaires) and in a larger cohort of patients.

The study population also lacked socioeconomic and racial diversity and the majority of participants were post-menopausal. Additionally, there was limited diversity in the BC subtype, as all participants presented with hormone receptor positive BC. Advanced BC stage was also underrepresented (4% with stage 3 BC). Given that research supports obesity is associated with multiple negative health sequalae in BC survivors regardless of receptor subtypes and stage [6,7,8,51], future studies should include a more diverse sample of these characteristics to ensure generalizability of the reported findings.

Future studies should investigate the effect of toxicities related to anti-cancer therapies on the uptake of healthy lifestyle behaviors and influence on DD. This is highly relevant in the modern era of breast cancer management given the rapid expansion of adjuvant treatments in cancer survivors to decrease recurrence risk; these treatments are not without side effects. We previously reported the adverse effects and associated severity related to endocrine therapy in this patient cohort [39]. The presence of aromatase inhibitor-induced musculoskeletal symptoms should be a particular area of future research, as this toxicity could impact physical activity and associated metabolic endpoints. Additionally, future studies should evaluate the unique psychological profile of individuals, including psychiatric diagnoses and psychologic support received following a cancer diagnosis. These factors may influence personal motivation to adhere to healthy lifestyle choices and an individual’s valuation of future health outcomes [52].

## 5. Conclusions

The current study suggests an association between DD and obesity in BC survivors and supports DD as a potential therapeutic target for weight loss. Additional research to elucidate the role of this behavioral mechanism in unhealthy lifestyle choices related to positive energy balance is indicated. Additionally, this relationship should be assessed in a more diverse BC survivorship cohort to confirm reproducibility. These findings support the potential for innovative behavioral therapies that target DD to promote weight loss and the associated long-term health benefits in BC survivors affected by increased weight. A randomized clinical trial by our research team implementing remotely delivered EFT intervention by smartphone is currently ongoing with these objectives (NCT05012176). If this strategy is feasible and efficacious to promote weight loss, improve diet quality, and reduce systemic inflammatory markers, additional investigation will be pursued across multiple cancer survivorship cohorts, as weight reduction in obesity is a pervasive target in optimal cancer survivorship care.

## Figures and Tables

**Figure 1 cancers-14-01134-f001:**
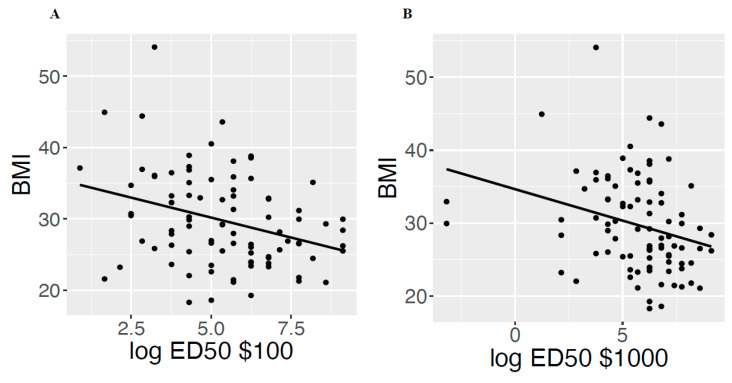
High discounting rate was independently associated with increased BMI (**A**): ED50 $100, *p* = 0.002; (**B**): ED50 $1000, *p* = 0.008).

**Table 1 cancers-14-01134-t001:** Baseline demographic, lifestyle, and clinical characteristics of study participants (n = 89).

Characteristics	Values	
Age (years)	mean (SD)	58.7 (10.2)
Sex	Female	89 (100%)
Race	n (%)	
Black or African American		2 (2.2)
White		85 (95.5)
Other		2 (2.2)
Ethnicity, not Hispanic or Latino	n (%)	88 (98.9)
Education	n (%)	
8th grade or less		1 (1.1)
High school graduate/GRE		11 (12.4)
Some college		28 (31.5)
College graduate		23 (25.8)
Graduate or professional degree		26 (29.2)
Income ($1000)	mean (SD)	95.6 (65.3)
Substance Use		
Tobacco use	n (%)	3 (3.4)
Drug use	n (%)	2 (2.2)
Alcohol consumption ^a^	Risk Level; n (%)	Low Risk; 89 (100%)
Menopausal status ^b^	n (%)	
Post-menopausal		58 (65.2)
Peri-menopausal		20 (22.5)
Pre-menopausal		11 (12.4)
BMI (kg/m²)	mean (SD)	29.8 (6.6)
Body Weight	n (%)	
Normal (BMI < 25 kg/m²)		22 (24.7)
Overweight (BMI 25 to <30 kg/m²)		29 (32.6)
Obese (BMI ≥ 30 kg/m²)		38 (42.7)
Breast Cancer Clinical Stage ^c^	n (%)	39 (43.8)
Stage 1		25 (28.1)
Stage 2		4 (4.5)
Stage 3		21 (23.6)
Unknown		
Time Since Diagnoses (years)	mean (SD)	2.5 (1.7)
Oral Endocrine Therapy	n (%)	53 (59.6)
Aromatase Inhibitor ^d^		36 (40.4)
Tamoxifen		
Vegetable Consumption (servings/day)	Median (IQR)	3 (2–4)
Fruit Consumption (servings/day)	Median (IQR)	3 (2–3)
Exercise Frequency	Median (IQR)	5 (4–6)
1 = never		
2 = less often than once a year		
3 = a few times a year		
4 = a few times a month		
5 = a few times a week		
6 = daily or almost daily		
Baseline Delay Discounting (log indifference delay)	Mean (SD)	
$100 ED50		5.38 (1.89)
$1000 ED50		5.69 (2.14)

^a^ Alcohol consumption based on Alcohol Use Disorders Identification Test (AUDIT) [41]; ^b^ Menopausal status was based on patient report; ^c^ Stage at diagnosis was obtained through medical records and was only available for patients recruited through physician referral (*n* = 68); ^d^ Aromatase inhibitor included anastrozole, letrozole, or exemestane; Abbreviation Legend: SD standard deviation; IQR interquartile range.

## Data Availability

The authors confirm that the data supporting the findings of this study are available within the article and/or the tables and figures. Additional derived data supporting the findings of this study are available from the corresponding author, Jennifer Vaughn MD, on request.

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
