# Peer review of "Delay Discounting as a Potential Therapeutic Target for Weight Loss in Breast Cancer Survivors"

_cancers, 2022, doi:10.3390/cancers14051134_

Round 1
Reviewer 1 Report
#1. The result table 1 can be revised into a more readable form, since right now, lots of variables are just listed in the center. Similar categories could be aligned in a row.
#2. Have you considered analyzing the assocations between DD and the patient's underlying disease (e.g. DM, hypertension, dyslipidemia) or social history (e.g. tobacco use, alcohol use)? These factors could also contribute to the result of this study.
Author Response
Response to Reviewer 1 Comments
Point 1: The result table 1 can be revised into a more readable form, since right now, lots of variables are just listed in the center. Similar categories could be aligned in a row.
Response 1: We thank reviewer 1 for the valuable comments. We have improved the table formatting to provide a more readable layout. We have adjusted the spacing and categorization of the headings and sub-headings. We have also included comments for the copy editors to ensure the correct alignment of the numerical values and statistical descriptions with the corresponding variables.
Point 2: Have you considered analyzing the associations between DD and the patient's underlying disease (e.g. DM, hypertension, dyslipidemia) or social history (e.g. tobacco use, alcohol use)? These factors could also contribute to the result of this study.
Response 2: We agree that cross-sectional analysis of other pertinent variables (e.g. underlying comorbidities, substance use) would provide additional insight into the relationship between delay discounting and lifestyle behaviors in this population of cancer survivors. However, our sample size is limited to provide any meaningful conclusions in this respect. Due to the small number of events, an exploratory analysis was not performed because we would not be able to portray an accurate representation of this correlation. Namely, there is wide variability regarding presence and type of metabolic comorbidities in this cohort. Additionally, substance use was uncommon as only three participants reported smoking tobacco, two participants reported drug use, and no patients engaged in risky alcohol consumption (based on the validated AUDIT score). Due to this relative rarity in the sample, covariance with substance use is unlikely to account for the study’s results. We have added descriptive information pertaining to substance use in table 1.
Reviewer 2 Report
The study addresses a relevant and important topic, with potential to be expanded in scope to impact cancer patient care. The limitations of the study and proposals for future to studies are well-discussed. Suggestions for further points of note to enhance the outreach -
- Patients on endocrine therapy may experience adverse effects, including musculoskeletal effects, that limit their healthy lifestyles and influence DD and BMI. Were these taken into account? It will be good also to discuss the impact of medications and treatment. 2
- Additionally, what was the psychological profile of the patients, were any of them treated for mental health issues related/not related to the diagnosis (eg. anxiety, depression)? Did patients have access to psychological support following diagnosis? Perspectives of the value of the future and motivation can be influenced by these factors. A discussion should be included.
- Were there changes in BMI from diagnosis to survey timepoint? This could give some insight into the evolution of the patients' responses to the cancer and facilitate the design of intervention tools. The change is important Eg. If overweight patients had been obese before, but actually lost weight after diagnosis, or vice versa, these have different implications. It would be good to discuss this.
Author Response
Response to Reviewer 2 Comments
The study addresses a relevant and important topic, with potential to be expanded in scope to impact cancer patient care. The limitations of the study and proposals for future to studies are well-discussed. Suggestions for further points of note to enhance the outreach -
Point 1: Patients on endocrine therapy may experience adverse effects, including musculoskeletal effects, that limit their healthy lifestyles and influence DD and BMI. Were these taken into account? It will be good also to discuss the impact of medications and treatment.
Response 1: We thank reviewer 2 for the insightful comments. We agree that certain adverse effects related to anticancer therapies may limit uptake of healthy lifestyle behaviors and could impact delay discounting. We have added this topic to the discussion as a future area of investigation (last paragraph):
“Future studies should investigate the effect of toxicities related to anti-cancer therapies on the uptake of healthy lifestyle behaviors and influence on DD. This is highly relevant in the modern era of breast cancer management given the rapid expansion of adjuvant treatments in cancer survivors to decrease recurrence risk; these treatments are not without side effects. We previously reported the adverse effects and associated severity related to endocrine therapy in this patient cohort [39]. The presence of aromatase inhibitor-induced musculoskeletal symptoms should be a particular area of future research as this toxicity could impact physical activity and associated metabolic endpoints.”
Point 2: Additionally, what was the psychological profile of the patients, were any of them treated for mental health issues related/not related to the diagnosis (eg. anxiety, depression)? Did patients have access to psychological support following diagnosis? Perspectives of the value of the future and motivation can be influenced by these factors. A discussion should be included.
Response 2: We agree with the reviewer’s comment concerning the potential impact of an individual’s psychological profile on delay discounting. Details surrounding psychologic support would certainly add to this study, however this data was not feasible to obtain as the sample of patients received heterogenous supportive services reflective of community setting care. We have incorporated the reviewer’s point in the last paragraph of the discussion:
“Additionally, future studies should evaluate the unique psychological profile of individuals, including psychiatric diagnoses and psychologic support received following a cancer diagnosis. These factors may influence personal motivation to adhere to healthy lifestyle choices and an individual’s valuation of future health outcomes [52]”
Point 3: Were there changes in BMI from diagnosis to survey timepoint? This could give some insight into the evolution of the patients' responses to the cancer and facilitate the design of intervention tools. The change is important Eg. If overweight patients had been obese before, but actually lost weight after diagnosis, or vice versa, these have different implications. It would be good to discuss this.
Response 3: This study was designed and powered to assess the relationship between endocrine therapy adherence and delay discounting. The cross-sectional investigation of obesity and delay discounting is exploratory; data regarding BMI collected serially over time is thus limited and would be challenging to accurately obtain. Changes in weight and body composition following a cancer diagnoses is not uncommon and can have important implications in the results of this study. An ongoing study by our research team investigating DD as a therapeutic target for weight loss in breast cancer survivors is collecting weight, anthropometric measures, and metabolic endpoints at several time points in a prospective manner (NCT05012176). The following has been added to the third paragraph of the discussion:
“Longitudinal assessment of adiposity collected serially would also provide a more comprehensive understanding of obesity and DD following a cancer diagnosis. Namely, changes in weight and body composition following a cancer diagnoses is not uncommon and can have important clinical implications. This additional data could inform the optimal design of behavioral interventions to target DD for weight loss. Our investigative team is currently evaluating the prospective effects of DD and related targeted interventions on weight loss after a breast cancer diagnosis (NCT05012176).”